# Accumulation of Labile P Forms and Promotion of Microbial Community Diversity in Mollisol with Long-Term Manure Fertilization

**Shuhui Song [1,2], Jinyao Zhang [1], Yunxia Liu [1] and Hong Wang [1,*]**

1 Key Laboratory of Plant Nutrition and Fertilizer, Ministry of Agriculture and Rural Affairs, Institute of Agricultural Resources and Regional Planning, Chinese Academy of Agricultural Sciences, Beijing 100081, China
2 Key Laboratory of Tropical Crops Nutrition of Hainan Province, South Subtropical Crops Research Institute of Chinese Academy of Tropical Agricultural Sciences, Zhanjiang 524091, China
* Correspondence: wanghong01@caas.cn

**Abstract:** Soil phosphorus (P) can be divided into inorganic P (Pi) and organic P (Po). Microorganisms play essential roles in soil P transformation. However, there are many ways to detect P transformation, and the relationship between P forms and microorganisms under long-term fertilization is largely unclear. In this study, soil P forms were analyzed by a chemical sequential fractionation method and solution $^{31}$P nuclear magnetic resonance ($^{31}$P-NMR) technique. Phospholipid fatty acid (PLFA) contents were measured by gas chromatography as the characterization of soil microbial community structures. The objective was to determine the changes of soil P forms and associated microbial community composition in mollisol with long-term fertilization. We sampled soil from a field experiment with 26-year-old continuous maize (*Zea mays* L.) cropping in Northeastern China. Three fertilization treatments were selected as chemical fertilization (NPK), NPK with crop straw (NPKS), and NPK with manure (NPKM). As shown in $^{31}$P-NMR spectra, orthophosphate accounted for 62.8–85.8% of total extract P. Comparison to NPK and NPKS treatments, NPKM application notably increased the concentrations of Po, Olsen-P, orthophosphate, orthophosphate monoester, and total P. Soil P fractions including resin-Pi, NaHCO$_3$-P, NaOH-P, and HCl-P, especially Pi fractions, were enhanced by NPKM. The amounts of total PLFAs and PLFAs in bacteria, Gram-positive (G$^+$) and Gram-negative (G$^-$) bacteria, actinomycetes, and fungi were high in NPKM-treated soil. The percentages of PLFAs in bacteria and fungi in total soil PLFAs were 56.8% and 9.7%, respectively, which did not show any significant difference among the treatments. NPKM increased the proportions (%) of PLFAs in G$^+$ bacteria, and NPKS increased the proportions (%) of G$^-$ bacteria in total PLFAs. The composition of soil microbial community was found to be significantly affected by soil total carbon and pH. There was a close relationship between HCl-Pi, NaHCO$_3$-Po, orthophosphate, and pyrophosphate with anaerobe, aerobes, and G$^+$. Manure addition directly increased soil available P concentrations, and indirectly acted through the alterations of anaerobe, aerobes, and G$^+$. It is concluded that long-term NPKM application would lead to the accumulation of labile P and moderately labile P in mollisol through the activity of soil microbes.

**Keywords:** long-term fertilization; mollisol soil; phosphorus forms; phospholipid fatty acid; $^{31}$P nuclear magnetic resonance; soil microbial community

## 1. Introduction

Phosphorus (P) is an essential macro-element for plant growth and plays a vital role in energy storage and transformation, cell membrane structure, cell reproduction, and gene expression. It acts as a component of biological macromolecules such as DNA, RNA, ATP, and phospholipids [1]. The various P forms and their transformation processes in the soils affect P availability for crop production. Soils' P is divided into inorganic P

(Pi) and organic P (Po) forms. The concentrations of total P in soils vary widely from 100 to 3000 mg·kg$^{-1}$ [2–4]. Pi generally contains orthophosphate, pyrophosphate, and polyphosphate. Orthophosphate as soluble orthophosphate anions ($H_2PO_4^-$ or $HPO_4^{2-}$) can be taken up directly by plant roots. However, the mobility of $H_2PO_4^-$ and $HPO_4^{2-}$ to plant roots in soil is low as orthophosphate is easy to be adsorbed and fixed by soil colloid [2,4]. Soil Po, comprising 30–65% of soil total P, can be absorbed by soil colloid particles or be mineralized to soluble Pi for plant utilization by soil microorganisms or soil enzymes [2–4]. Soil Po compounds are classified into orthophosphate monoesters and diesters, and phosphonates according to the specific P binding forms [2,4,5]. In order to overcome the low plant availability of P in soil and to meet the P demand for high crop yield, P fertilizers with reasonable rates are required to be applied to soils. Manure contains Pi and Po compounds. The Pi concentration in manures varies from 60% to 90% and is readily utilized by plants as it is soluble [2,6].

Sequential P fractionation method is a key technique to evaluate the availabilities of various P forms for plants and to detect the P transformation in soils [7,8]. The widely used procedure was developed by Hedley et al. (1982) [2] and then modified by Tiessen and Moir (1993) [9]. Guppy et al. (2000) [10] also proposed a modified procedure of Hedley's fractionation method to eliminate the centrifuge steps with ultra-high speed. The P extracted by anion exchange resin (Resin-P) and $NaHCO_3$ solution ($NaHCO_3$-P) is considered as the labile P form in soil that is readily taken up by plants [9]. The P forms extracted by NaOH (NaOH-P) and HCl (HCl-P) solution are identified to be bound with iron (Fe) and aluminum (Al) oxides, or as Ca-P compounds [11]. Residual P is a nonlabile and stable form that is less available to plants [12].

One-dimensional solution $^{31}$P nuclear magnetic resonance spectroscopy ($^{31}$P-NMR) is commonly used to identify Po forms in soils [5,13,14] and sediments [15,16]. Mixture of ethylenediaminetetraacetate (EDTA) and NaOH is commonly used as an extract solution of soil Po [5,13,14]. More details of the technologies using solution $^{31}$P NMR for soils include preparing samples, and NMR experiment parameters can be found in the reviews by Cade-Menun and Liu (2014) [13] and Cade-Menun et al. (2017) [14]. Soil P compounds were often identified as orthophosphate, orthophosphate monoesters, and orthophosphate diesters based on the data report of the $^{31}$P-NMR method [14].

Microorganisms play a critical role in P transformation and cycle in soils as they are involved in solubilization, mineralization, and immobilization of soil P [17,18]. Certain microorganisms can dissolve soil Pi from insoluble and fixed/adsorbed forms into soluble Pi form available for plant uptake, by reducing soil pH, releasing organic acids, and increasing the activities of chelating with P adsorption sites [19]. The breakdown process of soil Po compounds to Pi is catalyzed by phosphatases released from soil microorganisms or plant roots [20]. Phospholipid fatty acids (PLFAs) are used as biomarkers for soil microorganism community structure [21,22] as PLFAs are important components in the cell membranes of all soil microbes. Polar head groups and ester-linked side chains of PLFAs show differently in cell membranes in the compositions of eukaryotes and prokaryotes, as well as vary in many prokaryotic groups [21–23].

Long-term experiment in the field is an important resource for investigating the dynamics of soil nutrients. Koopmans et al. (2007) [24] used solution $^{31}$P-NMR to determine NaOH-EDTA extractable P forms in the noncalcareous sandy soil in the Netherlands applied for 11 years with excessive rates of pig slurry, poultry manure, or poultry manure mixed with litter. They found that orthophosphate and orthophosphate monoesters accumulated in heavily manured topsoils, suggesting that Po accumulation is likely to be associated with high organic C in soil. Zhang et al. (2014) [25] revealed that orthophosphate and phosphomonoesters accounted for 59–84% and 15–40% of P composition in the NaOH-EDTA extracts, respectively. Compared with chemical fertilizer addition, application of manure for 25 years increased soil Po, especially *myo*-inositol hexakisphosphate (*myo*-IP). Using solution $^{31}$P-NMR technique, Abdi et al. (2014) [26] demonstrated accumulation of orthophosphate monoesters, IP, and nucleotides in the deep layer of soils with long-term

no till treatment, which was possibly due to preferential P movement to the deep layer. Annaheim et al. (2015) [27] found the concentrations of *scyllo*-IP, *myo*-IP, pyrophosphate, and degradation products from phospholipids and nucleic acids were similar among the treatments with three organic fertilizers including dairy manure, compost, and sewage sludge. It suggested that soil Po compositions did not significantly change under long-term applications of organic fertilizers compared with conventional mineral fertilizer additions. Liu et al. (2019) [28] used the solution $^{31}$P-NMR technique to study the molecular speciation of P in an alkaline soil with or without mineral P fertilizers in a wheat/maize rotation system for 26 years. Results showed orthophosphate monoesters increased with P fertilization; however, this P form did not significantly decline when withholding P fertilization. Xin et al. (2019) [29] classified P forms in fluvo-aquic soil with 26 years of different fertilization treatments including full usage of organic compost, 1/2 compost +1/2 NPK (mineral fertilizer), NPK, NK, and control of unfertilization. Results indicated that orthophosphate accounted for 64.3–83.5% of the total P extracted with NaOH-EDTA solution. The addition of P fertilizer significantly increased the concentrations of orthophosphate, orthophosphate monoesters, and orthophosphate diesters regardless of P treatments. The proportions and concentrations of orthophosphate significantly decreased under compost application. It can be seen that P forms changed after long-term manure fertilization.

In this study, we sampled soils from a 26-year-old continuous maize (*Zea mays* L.) cropping field experiment in Northeastern China. Three fertilization treatments were selected as chemical fertilization (NPK), combined application of NPK with crop straw (NPKS) and combined application of NPK with manure (NPKM). The objectives of the present research were (a) to study the change of soil P pools and their dynamics determined using $^{31}$P NMR spectroscopy and chemical sequential fractionation method under the condition of long-term applications; (b) to characterize soil microorganisms community structures and their P patterns (PLFA, P-lipid quantification) as affected by long-term fertilization; and (c) to assess the relationship between changes in soil P pools and soil microbial community.

## 2. Materials and Methods

### 2.1. Filed Experiment

The experiment has been conducted since 1990 at Gongzhuling City of Jilin Province in northeastern China (43°30′23″ N, 124°48′34″ E). The mean annual rainfall is 550 mm, and the annual average temperature is 4.5 °C. The soil at the experiment site is classified as Mollisol based on the USDA soil taxonomy system [30].

The properties of original soil (0–20 cm) were 7.60 of pH value, 22.80 g·kg$^{-1}$ of organic matter, 1.40 g·kg$^{-1}$ of total N, 18.42 g·kg$^{-1}$ of total K, 0.61 g·kg$^{-1}$ of total P, and 11.79 mg·kg$^{-1}$ of Olsen-P. The cropping system is continuous maize throughout the experiment.

The three fertilization treatments were selected: (1) chemical fertilization (NPK), with an annual rate of 165 kg N ha$^{-1}$ as urea, 82.5 kg P$_2$O$_5$ ha$^{-1}$ as triple superphosphate, and 82.5 kg K$_2$O ha$^{-1}$ as potassium sulfate; (2) manure with 30 t·ha$^{-1}$ + NPK (NPKM). Manure was used as pig manure before 2004 and cow manure after 2005. On average, pig manure had 51.5 kg P ha$^{-1}$ and cow manure had 28.5 kg P ha$^{-1}$; and (3) straw at 7.5 t·ha$^{-1}$ +NPK (NPKS). Each plot was 400 m$^2$. One-third of N and all PK mineral fertilizers were basally added into soil. A second third of N was used as a supplemental fertilizer at jointing stage. Manure and straw were broadcasted and incorporated into the soil in late October after maize plants were harvested each year.

### 2.2. Soil Samples and Chemical Analysis

Soils in the layer from 0 to 20 cm deep were sampled in 2016 after the crop harvest. The mean annual rainfall is 565 mm and the annual average temperature was 6.8 °C from 1990 to 2016. Three replicate samples per treatment were collected with 10 sample soil cores

each replicate. Some fresh samples were sieved through 2 mm mesh and stored at $-80\,°C$. The other soil samples were air-dried, sieved, and stored prior to analysis.

Soil total C and total N were analyzed using an elemental analyzer (vario PYRO cube, Elementar Analysensysteme GmbH, Shanghai, Germany). Soil total P was digested with $H_2SO_4$-$HClO_4$ and determined by molybdenum-blue colorimetry method [31,32].

Soil pH at a soil-to-water ratio of 1:2.5 was detected with a pH meter with glass electrode. Soil Olsen-P was extracted with $0.5\ mol·L^{-1}$ NaHCO$_3$. Soil Po was calculated by the difference of Pi concentration in $0.5\ mol·L^{-1}$ of $H_2SO_4$ extract solution between ignited $(550\,°C, 1\ h)$ and unignited soils [32]. The concentration of Pi in all extracts was determined using molybdenum-blue colorimetry method [31,32].

### 2.3. NMR Experiments

Soil samples were determined by solution $^{31}P$-NMR spectroscopy based on the procedure of Cade-Menun and Preston (1996) [13]. Next, 2.5 g of air-dried soil through a 2 mm sieve was treated with 50 mL of extraction solution containing $0.25\ mol·L^{-1}$ NaOH and $50\ mmol·L^{-1}$ Na$_2$EDTA and shaken for 16 h at $25\,°C$ in a temperature-controlled shaker (HZ-9212S, Jiangsu Taicang Scientific equipment factory, Taicang, China). The extracts were then centrifuged for 30 min at $10,000\times g$ at $4\,°C$. The supernatant was added with 1.5 mL methylenediphosphonic acid $(50\ \mu g\ P·mL^{-1})$ as a calibration compound and subsequently lyophilized. Then, 200 mg of lyophilized material was re-dissolved into 1 mL of $1\ mol·L^{-1}$ NaOH and $50\ mmol·L^{-1}$ Na$_2$EDTA, and 0.1 mL of deuterium oxide (D$_2$O) was spiked to tune the NMR instrument. The solution was centrifuged at $10,000\times g$ for 8 min at $4\,°C$, 800 $\mu$L of supernatant was transferred to a 5 mm diameter NMR tube. $^{31}P$-NMR spectra were measured at 298 K on a Bruker DRX-400 NMR Spectrometer (Bruker BioSpin GmbH, Rheinstetten, Germany). The parameters in the NMR experiment were as follows: pulse width of 14 $\mu$s $(90°)$, acquisition 1.8 s, delay time 2 s, 20,480 scans, no proton decoupling.

The spectra processing was performed using software of TopSpin 3.5 (Bruker BioSpin GmbH, Rheinstetten, Germany). $^{31}P$ chemical shifts $(\delta)$ of signals were expressed in parts per million (ppm) relative to an external standard (at $\delta$ 0.0 ppm) with 85% orthophosphoric acid. Signals in the spectrum for soil P compounds were assigned based on the literature [26,33,34]. The orthophosphate peak was at $\delta$ 6 ppm.

The relative proportion of each soil P compound was calculated by integration on spectra processed with 2 and 7 Hz line broadening. The absolute concentrations of the different P forms were obtained by comparing their integrals with that of the calibration compound methylenediphosphonic acid.

### 2.4. Sequential Fractionation

A sequential extraction method proposed by Guppy et al. (2000) [10] was used to successively identify five P fractions in soil. Then, 0.5 g of soil sample was extracted with a specific solution added to a centrifuge tube with 100 mL volume in the following sequential order: (1) A total of 30 mL distilled water with two bags (1 g) of anion exchange resin (Resin-Pi and Resin-Po). (2) A total of 30 mL $0.5\ mol·L^{-1}$ NaHCO$_3$ solution at pH 8.5 (NaHCO$_3$-Pi and NaHCO$_3$-Po). (3) A total of 30 mL $0.1\ mol·L^{-1}$ NaOH solution and 1 mL $4\ mol·L^{-1}$ NaCl solution (NaOH-Pi and NaOH-Po). (4) A total of 30 mL $1\ mol·L^{-1}$ HCl solution (HCl-Pi and HCl-Po). (5) Residue soil was digested with 5 mL $H_2SO_4$:$HClO_4$ at a ratio of volume of 20:1 for 3 h (Residual-P). The tube was shaken for 16 h at 180 rpm at $25\,°C$ in a temperature-controlled shaker (HZ-9212S, Jiangsu Taicang Scientific equipment factory, Taicang, China). The extracts in the tube were centrifuged at 900 $g$ for 30 min and the supernatants were collected for P determination. The extracting process was repeated for each extractant including Resin, NaHCO$_3$, NaOH, and HCl. Then, 30 mL of $0.5·mol\ L^{-1}$ NaCl solution was used to elute phosphate anions adsorbed on the resin in the process of resin P extraction by shaking 1 h. P concentrations for all the extracts were determined using molybdenum-blue colorimetry method [31,32]. The total P concentration in the different extracts was determined by $H_2SO_4$-$HClO_4$ digestion. Po concentrations were

estimated as the difference between Pt and Pi. Based on the availability for plant uptake, resin P and $NaHCO_3$-P fractions were regarded as labile P, NaOH-P as moderately labile P, and HCl-P and residual P as non-labile or stable P pools [2,12].

### 2.5. Soil PLFA Analysis

The contents of PLFAs in soil were determined according to the procedure described by Wu et al. (2009) [35]. First, 2.50 g of freeze-dried soil samples were extracted with 15.2 mL of a mixture solution containing chloroform: methanol: citrate at a volume ratio of 1:2:0.8. The phospholipids in soil extract were methylated to form PLFA methyl esters, which were isolated and quantified using a gas chromatograph (GC) (Agilent 6890A, Agilent Technologies, Inc., Santa Clara, CA, USA) equipped with a flame-ionization detector and a capillary column of 30 m × 0.32 mm × 0.25 μm (HP-5, Agilent J&W Scientific, Folsom, CA, USA). Prior to the GC analysis, the samples were dissolved in 150 μL of pure hexane. Nonadecanoic acid methyl ester (19:0, Sigma-Aldrich, St. Louis, MO, USA) was used as an internal standard. The peaks of various PLFAs were identified by comparing the retention time with the standard compounds (MIDI Sherlock software system, Newark, DE, USA). The concentration of PLFA in soil was expressed as $nmol \cdot g^{-1}$ in a unit.

Soil PLFAs were categorized into different taxonomic groups according to the data previously published by Vestal and White (1989) [36], Frostegård and Bååth (1996) [21], and Kaiser et al., (2010) [22]. The fatty acids including 16:0, 17:0, 16:1ω5c, 16:1ω7c, 16:1ω9c, 17:1ω8c, 18:1ω5c, 18:1ω7c, anteiso15:0, anteiso17:0, cy17:0, cy19:0ω8c, iso14:0, iso15:0, iso16:0, iso17:0, and iso19:0 of bacterial origin were classified as biomarkers of bacterial biomass. We used fatty acids of iso14:0, iso15:0, iso16:0, iso17:0, anteiso15:0, and anteiso17:0 to represent Gram-positive ($G^+$) bacteria and used cy17:0, 16:1ω7c, 16:1ω9c, 17:1ω8c, 18:1ω5c, 18:1ω7c, and cy19:0ω8c to represent Gram-negative ($G^-$) bacteria. Total bacterial PLFAs was the sum of $G^+$ and $G^-$, and general bacterial PLFAs. The fatty acids 16:1ω5c, 18:2ω6,9c, and 18:1ω9c were used as biomarkers of fungal biomass, and 16:1ω5c was used as an indicator of arbuscular mycorrhizal fungi (AMF) biomass. The fatty acids16:0Me, 17:0Me, and 18:0Me were regarded as actinobacteria biomarkers. The total microbial lipid biomass was represented with the sum of all PLFAs in soil.

### 2.6. Statistics

A one-way analysis of variance (ANOVA) and a least significant difference (LSD) were used to test the differences among treatments at a 5% significance level ($p < 0.05$). CANOCO 4.5 software (Wageningen UR) was used to perform principal component analysis (PCA) and redundancy analysis (RDA), which would explore the relationship between microbial community with soil properties and soil P forms. Pearson correlation analysis was simultaneously performed within the measured parameters.

## 3. Results

### 3.1. Soil Characteristics and Crop Yield

Soil pH, total C, and total N were higher under the treatments of NPKS and NPKM than that under NPK. Soil with NPK application had the lowest total P, while soil with NPKM treatment was the highest. Soil with NPKM application showed the highest levels of Olsen-P and soil Po. The grain yield of maize did not show a significant difference between NPK and NPKS treatments, while the yield of NPKM treatment was greater than that of NPK and NPKS (Table 1).

**Table 1.** Soil characteristics in the layer of 0–20 cm deep and maize grain yield under long-term fertilization application.

| Treatments | pH | Total C (g·kg$^{-1}$) | Total N (g·kg$^{-1}$) | Total P (mg·kg$^{-1}$) | Olsen-P (mg·kg$^{-1}$) | Organic P (mg·kg$^{-1}$) | Grain Yield (kg·ha$^{-1}$) |
|---|---|---|---|---|---|---|---|
| NPK | 6.2 ± 0.1 c | 15.15 ± 0.19 c | 1.43 ± 0.04 c | 659.53 ± 1.92 c | 29.29 ± 1.47 b | 213.75 ± 0.55 b | 11,789.1 ± 338.0 b |
| NPKS | 7.5 ± 0.0 b | 17.45 ± 1.05 b | 1.55 ± 0.02 b | 717.56 ± 11.53 b | 18.16 ± 1.45 c | 214.81 ± 4.61 b | 11,464.3 ± 575.7 b |
| NPKM | 7.9 ± 0.3 a | 23.62 ± 0.46 a | 2.23 ± 0.05 a | 1628.98 ± 16.35 a | 131.6 ± 1.18 a | 323.15 ± 6.14 a | 12,938.2 ± 170.6 a |

Notes: Each value represents the mean ± standard deviation. The difference among the values with the same letter within one column are not statistically significant at the $p < 0.05$ level.

### 3.2. P Determination by 1D $^{31}$P-NMR Spectroscopy

Figure 1 showed the spectra of 1D $^{31}$P-NMR of the soils with treatments of NPK, NPKS and NPKM. The chemical shift ranged from 20 to −10 ppm. The peaks of orthophosphate and pyrophosphate were at 6.00 ppm and −3.50–5.00 ppm, respectively. Peaks of orthophosphate monoester were observed in the range of 2.57 to 6.00 ppm. The peaks of orthophosphate diesters were at 2.50 ppm to −0.1 ppm. Signal of 17.68 ppm was methylenediphosphonic acid as calibration compound. Orthophosphate, pyrophosphate, and orthophosphate monoester compounds were found under the treatments of NPK, NPKS, and NPKM.

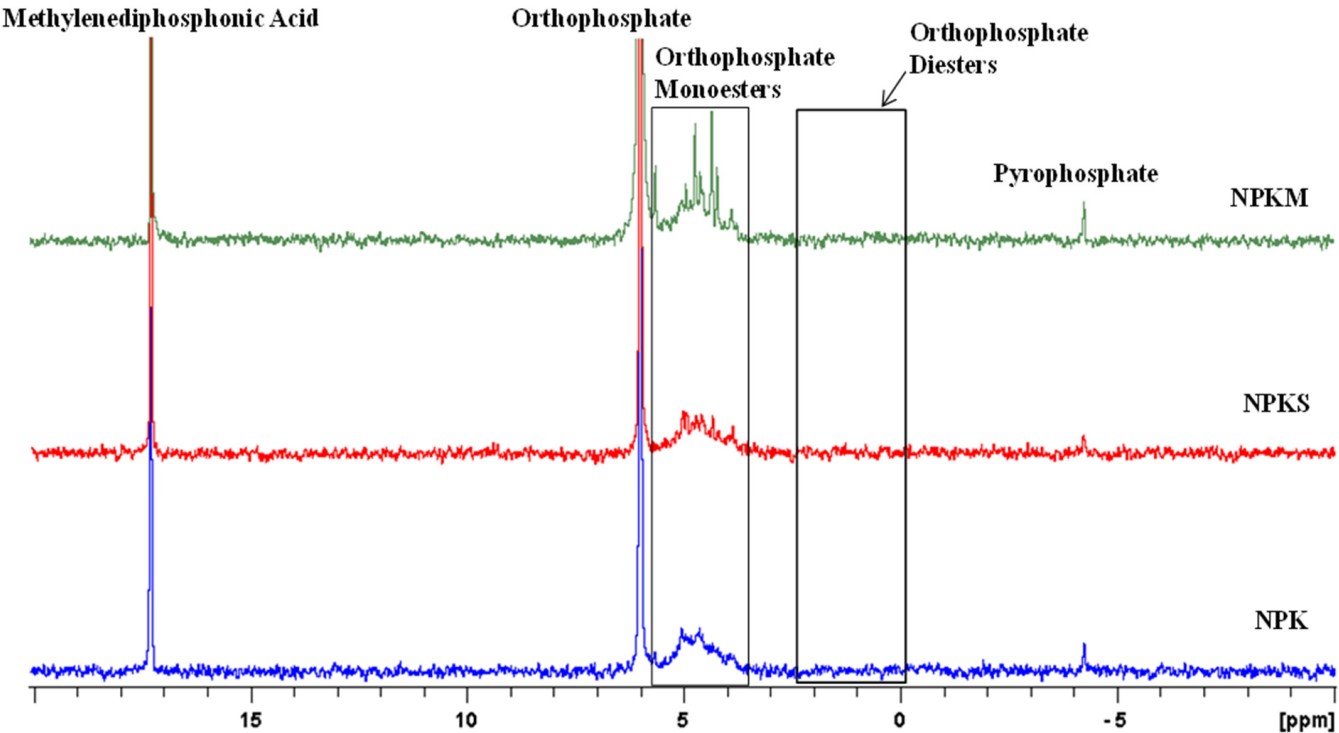

**Figure 1.** Solution $^{31}$P-NMR spectra of mollisol under long−term different fertilizer treatments.

The expanded spectra of the orthophosphate monoester region at the chemical shift of 3 to 6 ppm were shown in Figure 2. The peaks of 3.11 to 3.32 ppm(d) and 3.85 to 3.87 ppm(b) were sugar phosphates and *scyllo*-IP, respectively. The peak of 4.09 ppm(g) was an unknown compound. The peaks of 4.55 to 5.03 ppm(e,f) and 4.45 to 4.65 ppm(h) in the spectra were IP and orthophosphate diester degradation products. In the spectra of NPKM-treated soil, the peaks at 5.65(c1), 4.73(c2), 4.35(c3), and 4.22 ppm(c4) were more oblivious than those of NPK and NPKS, which were identified as *myo*-IP in the typical 1:2:2:1 configuration.

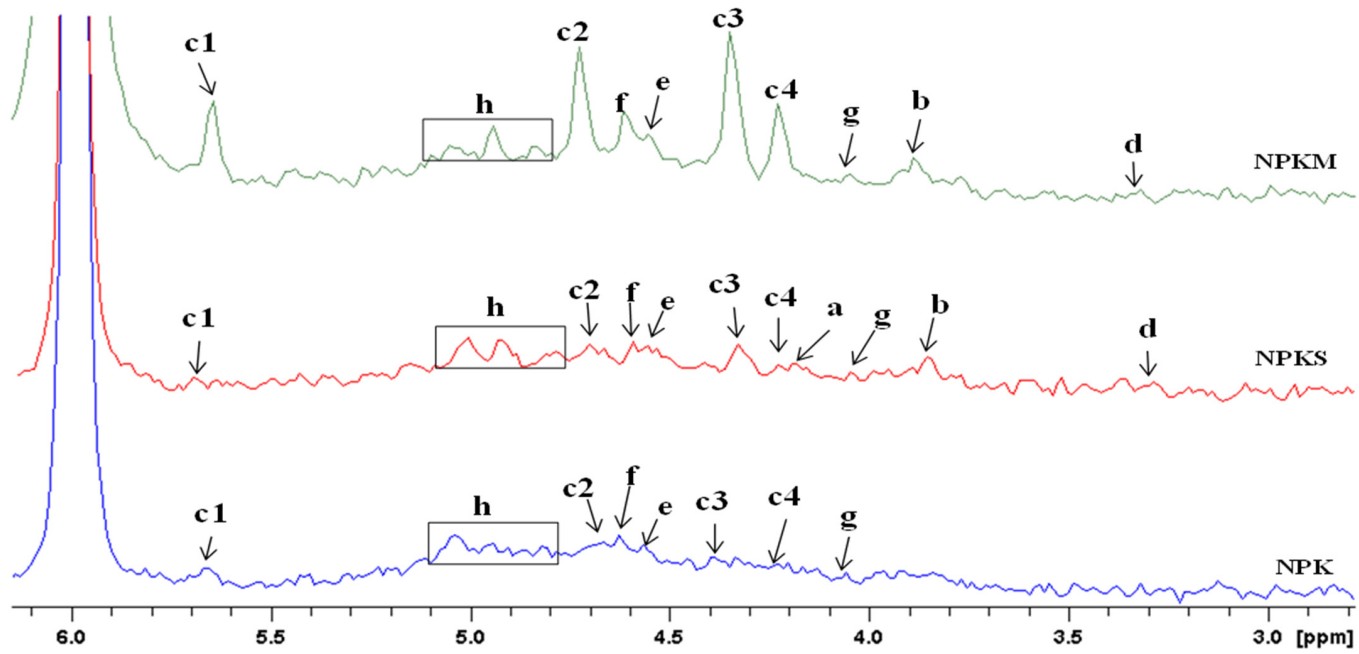

**Figure 2.** Solution $^{31}$P-NMR spectra of mollisol under long−term different fertilizer treatments showing orthophosphate monoesters region in detail (7.0–3.0 ppm). a: β−glycerophosphate; b, d, g: sugar phosphates or other unknown monoesters; c1, c2, c3, c4: *myo*−inositol hexakisphosphate (IP); e, f, h: IP and orthophosphate diester degradation products.

Table 2 showed the percentages and concentrations of various P compounds in soil with NaOH-EDTA extracts based on 1D $^{31}$P-NMR spectra. The majority of P was orthophosphate, accounting for 62.8–85.8% of the total extracted P. The proportion of pyrophosphate was less than 2%. Compared with NPK and NPKS treatments, NPKM application significantly enhanced the concentrations of orthophosphate and orthophosphate monoester in soil. The percentages of orthophosphate and orthophosphate monoester in the extractable total P of NPKM treatment were 85.8% and 12.8%, respectively. The proportion of orthophosphate and orthophosphate monoester in total extractable P of NPK treatment accounted for 62.8% and 32.1%, respectively. The proportion of orthophosphate and orthophosphate monoester in total extractable P of NPKS treatment was 67.3% and 29.0%, respectively. The percentage (1.0%) of orthophosphate diesters in total P of soils under NPKM fertilization was remarkably lower than that of NPK and NPKS.

**Table 2.** Concentration (mg·kg$^{-1}$) and proportions (%) of various P compounds identified by 1D $^{31}$P-NMR technique in the NaOH–EDTA extracts of soils under long-term fertilization.

| Treatments | Orthophosphate | Orthophosphate Monoesters | Orthophosphate Diesters | Pyrophosphate | Total Extract P |
|---|---|---|---|---|---|
| NPK | 187.6 ± 2.9 c (62.8 ± 1.1 c) | 95.9 ± 1.3 b (32.1 ± 0.3 a) | 8.2 ± 1.2 b (2.7 ± 0.4 a) | 7.0 ± 1.4 a (2.3 ± 0.5 a) | 298.7 ± 1.2 c |
| NPKS | 230.2 ± 3.1 b (67.3 ± 1.4 b) | 99.2 ± 4.0 b (29.0 ± 0.3 b) | 9.0 ± 3.1 ab (2.6 ± 0.8 a) | 3.6 ± 1.1 b (1.1 ± 0.3 b) | 342.2 ± 9.7 b |
| NPKM | 1088.3 ± 1.6 a (85.8 ± 0.2 a) | 161.9 ± 2.9 a (12.8 ± 0.2 c) | 12.4 ± 0.4 a (1.0 ± 0.0 b) | 6.0 ± 0.5 a (0.5 ± 0.0 b) | 1268.6 ± 1.0 a |

Notes: Each value of P concentrations represents the mean ± standard deviation (*n* = 3). Concentrations of various P compounds were quantified according to the peak area of $^{31}$P-NMR spectrum in Figure 1 by comparing the integrals of interest to that of the calibration compound methylenediphosphonic acid (50 μg P mL$^{-1}$). The values in parentheses are the proportions (%) of the total P assigned to each P compound. Different lowercase letters indicate significant difference among the treatments based on LSD test (*p* < 0.05).

### 3.3. P Fractionation in Soil

The dominant P fraction of the soil was HCl-P, which was 36~39% of the total P, followed by NaOH-P, which was 22~30% of the total P. Resin-P and NaHCO$_3$-P made up 19~32% of the total P. Residual P was the smallest fraction, accounting for 6~15% of total P (Table 3).

**Table 3.** Concentration (mg·kg$^{-1}$) and proportions (%) of various P forms determined using sequential extraction method in soils under long-term fertilization.

| Treatments | Resin-Pi | Resin-Po | Resin-P | NaHCO$_3$-Pi | NaHCO$_3$-Po | NaHCO$_3$-P | NaOH-Pi | NaOH-Po | NaOH-P | HCl-Pi | HCl-Po | HCl-P | Residual-P |
|---|---|---|---|---|---|---|---|---|---|---|---|---|---|
| NPK | 32.0 ± 3.7 b | 13.4 ± 2.9 a | 45.4 ± 0.9 b | 49.9 ± 2.3 b | 27.2 ± 6.9 b | 77.1 ± 4.6 b | 90.0 ± 8.0 b | 100.3 ± 14.9 b | 190.4 ± 7.3 b | 138.3 ± 3.0 c | 89.8 ± 11.9 b | 228.1 ± 10.9 c | 92.2 ± 2.2 b |
| NPKS | 36.2 ± 1.0 b | 13.4 ± 1.1 a | 49.6 ± 1.2 b | 44.4 ± 2.8 b | 38.5 ± 2.7 b | 82.9 ± 4.0 b | 98.5 ± 3.4 b | 97.3 ± 4.5 b | 195.9 ± 1.1 b | 173.9 ± 7.2 b | 83.1 ± 8.7 b | 257.1 ± 9.4 b | 102.2 ± 4.3 a |
| NPKM | 199.3 ± 2.0 a | 11.2 ± 3.8 a | 210.5 ± 4.2 a | 261.7 ± 5.3 a | 56.9 ± 19.1 a | 318.5 ± 15.0 a | 210.6 ± 5.1 a | 152.8 ± 1.2 a | 363.4 ± 5.1 a | 527.8 ± 11.1 a | 116.7 ± 2.6 a | 644.5 ± 12.1 a | 108.3 ± 4.9 a |

Notes: Each value of P concentrations represents the mean ± standard deviation ($n = 3$). The values in parentheses are the proportions (%) of the total P assigned to various P forms. Different lowercase letters indicate significant difference among the treatments based on LSD test ($p < 0.05$).

There was no significant difference in the concentrations of various soil P forms between the treatments of NPK and NPKS. In comparison with NPK and NPKS treatments, NPKM application notably increased Resin-Pi, NaHCO$_3$-Pi and Po, NaOH-Pi and Po, HCl-Pi and Po. Moreover, NPKM treatment increased more Pi fractions in soil than Po.

### 3.4. Soil PLFAs

The contents of total PLFAs and PLFAs in bacteria, G$^+$ bacteria, G$^-$ bacteria, actinomycetes, aerobes, AMF, and fungi in NPKM-treated soil were the highest, followed by NPKS treatment, and NPK applied soil had the lowest value (Table 4). The proportions (%) of PLFAs in various soil microorganisms to total PLFAs did not show obvious difference between NPKM and NPK treatments. Compared to the NPK, NPKS appeared to have higher abundances of actinomycetes and G$^-$ bacteria, and higher proportions (%) of PLFAs in actinomycetes and G$^-$ bacteria to total soil PLFAs.

**Table 4.** Amounts (nmol g$^{-1}$) and percentage (%) of phospholipid fatty acids (PLFAs) in soil microorganisms to total PLFAs in soil under long-term fertilizer treatments.

| | Actinomycetes | G$^+$ | G$^-$ | Fungal | Bacteria | AMF | Aerobes | Anaerober | Total PLFA |
|---|---|---|---|---|---|---|---|---|---|
| NPK | 5.0 ± 0.1 c (12.4 ± 0.0 b) | 8.8 ± 0.2 b (21.9 ± 0.0 a) | 11.0 ± 0.2 c (27.3 ± 0.0 a) | 3.7 ± 0.1 c (9.2 ± 0.0 b) | 22.1 ± 0.8 b (55.0 ± 2.9 a) | 0.9 ± 0.1 c (2.3 ± 0.2 b) | 2.8 ± 0.2 c (7.0 ± 0.5 a) | 4.6 ± 0.5 b (11.3 ± 1.3 a) | 40.2 ±0.8 c |
| NPKS | 7.8 ± 0.2 b (13.6 ± 1.1 ab) | 9.8 ± 0.1 b (21.9 ± 0.3 a) | 16.6 ± 0.5 b (26.7 ± 3.7 a) | 5.2 ± 0.1 b (9.8 ± 0.4 a) | 30.8 ± 0.5 b (55.2 ± 7.2 a) | 1.8 ± 0.1 b (3.0 ± 0.4 a) | 3.5 ± 0.2 b (7.3 ± 1.0 a) | 4.7 ± 0.2 b (10.5 ± 2.1 a) | 53.9 ±1.0 b |
| NPKM | 11.7 ± 0.3 a (14.5 ± 0.0 a) | 18.9 ± 1.5 a (18.1 ± 0.2 b) | 23.2 ± 4.5 a (30.7 ± 0.4 a) | 8.4 ± 0.2 a (9.6 ± 0.0 ab) | 47.9 ± 8.9 a (57.2 ± 0.1 a) | 2.5 ± 0.2 a (3.3 ± 0.2 a) | 6.3 ± 0.5 a (6.5 ± 0.4 a) | 9.0 ± 1.2 a (8.7 ± 0.4 a) | 86.4 ±5.5 a |

Notes: Each value of P concentrations represents the mean ± standard deviation ($n = 3$). The values in parentheses are the proportions (%) of phospholipid fatty acids (PLFAs) in various soil microorganisms to total soil PLFA. Different lowercase letters indicate significant difference among the treatments based on LSD test ($p < 0.05$). AMF, Arbuscular mycorrhizal fungi.

The results of the PCA indicated that there were notable effects of the different fertilization treatments on PLFA compositions. Figure 3 showed the results of loading values for each biomarker of soil PLFAs under the different treatments. The first principal component (PC1) made up 65.7% and the second component (PC2) made up 33.9% of the variation in the PCA profiles of PLFA contents. The biomarkers related to G$^+$ bacteria, such as iso 16:0 and anteiso 17:0, the biomarkers associated with G$^-$ bacteria, including cy17:0, 17:1ω8c, and 18:1ω7c, and the general bacteria biomarkers, namely cy17:0, 16:1ω5c, and iso 19:0, were more abundant under the NPKM treatment than that under NPKS and NPK. NPKM application increased the concentrations of 16:1ω5c, 18:2ω6,9c, and 18:1ω9c, representing fungal PLFA biomarkers. The loading values of two PLFA, namely 16:0Me and 17:0 Me,

representing actinomycete biomarkers were most important for the NPKM treatment, as the relative abundance was the highest in this treatment. The two PLFA biomarkers associated with G$^-$ bacteria, 18:1$\omega$5c and 16:1$\omega$9c, were abundant under the NPKS and NPK treatment.

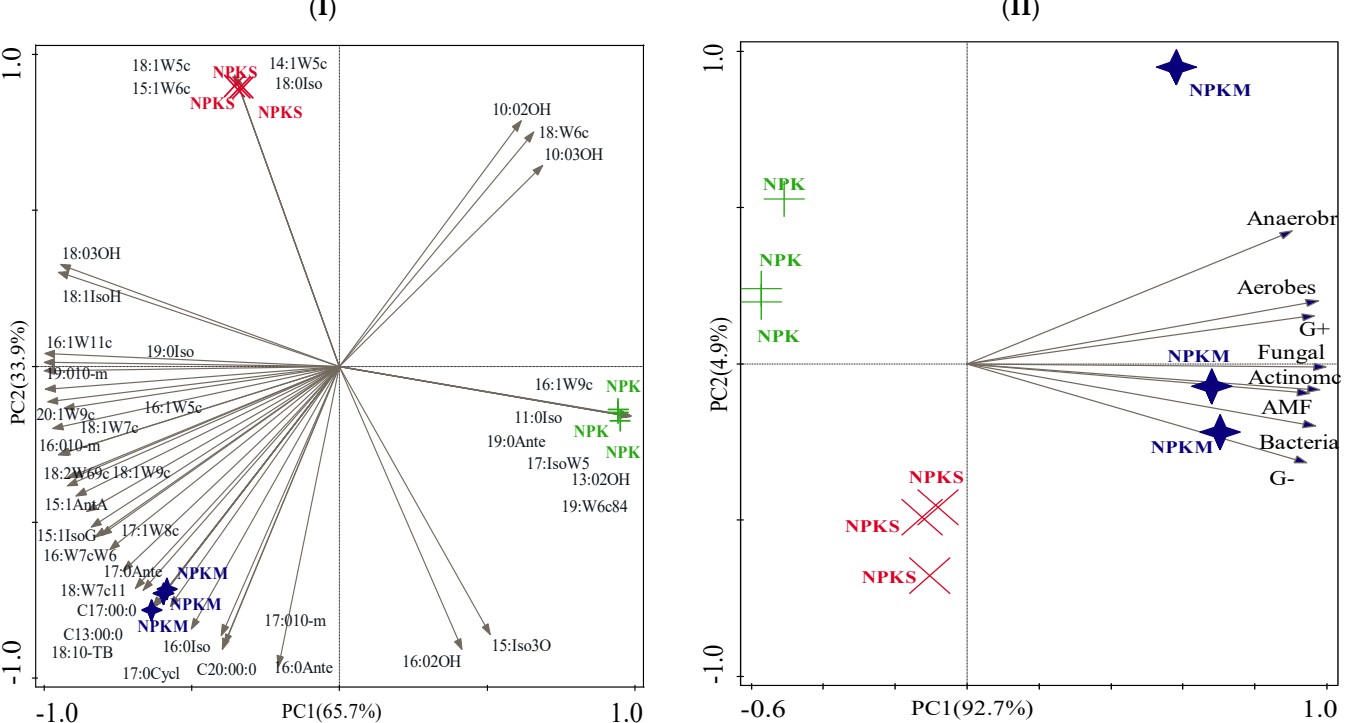

**Figure 3.** Principal component analysis (PCA) of the contents of PLFAs and microbial community structure in soil across different long−term fertilization. (**I**) Loading values of various PLFAs. (**II**) Loading values of soil microbial community.

The microbial community composition between different treatment was separated via PCA mainly on PC1 (92.7%) (Figure 3). Three different fertilization treatments were clearly distinguished, suggesting significant differences of soil microbial community structure occurred among different fertilization treatments. NPKM treatment was at the positive end of PC1, while NPKS was at the negative. It was observed that PLFA biomarkers were most important for the NPKM treatment with the highest relative abundance of microbial community composition, suggesting NPKM application obviously promoted the diversity of soil microbial community structure.

*3.5. Relationship between Selected Soil Properties and Microbial Community Structure*

The RDA results revealed soil properties including TC (89.0%, F = 56.5, *p* = 0.004) and pH (6.9%, F = 9.9, *p* = 0.002) strongly affected the composition of soil microbial community, indicating that TC and pH played significant roles in shaping the different soil microbial communities under long-term fertilization (Figure 4). Considering the variables of P forms, HCl-Pi (84.5%, *F* = 38.3, *p* = 0.004) and NaHCO$_3$-Po (4.7%, *F* = 5.6, *p* = 0.038) significantly affected the microbial community composition (Figure 5). Orthophosphate, orthophosphate diesters, and orthophosphate monoesters highly corrected with aerobes, anaerobes, and G$^+$. Orthophosphate and pyrophosphate explain the variability of microbial community structure 82.8% and 9.6%, respectively (Figure 6), indicating NPKM increased soil orthophosphate, orthophosphate diesters, and orthophosphate monoesters, which might stimulate soil microbial structure, especially anaerobe, aerobes, and G$^+$.

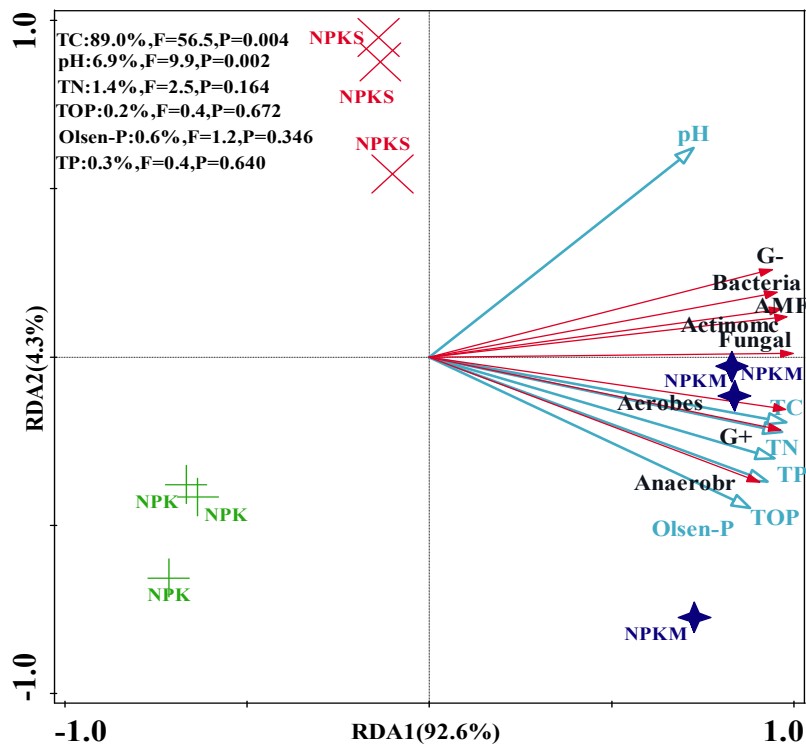

**Figure 4.** Redundancy analysis was performed on soil microbial community structure marked by PLFA profiles and selected soil properties under different long−term fertilization. PLFA profiles are indicated by red arrows. G$^+$ means Gram−positive bacterial PLFAs; G$^-$ means gram−negative bacterial PLFAs. Soil variables are showed as blue arrow: total soil phosphorus (TP), total soil organic phosphorus (TOP), total carbon (TC), and total soil nitrogen (TN).

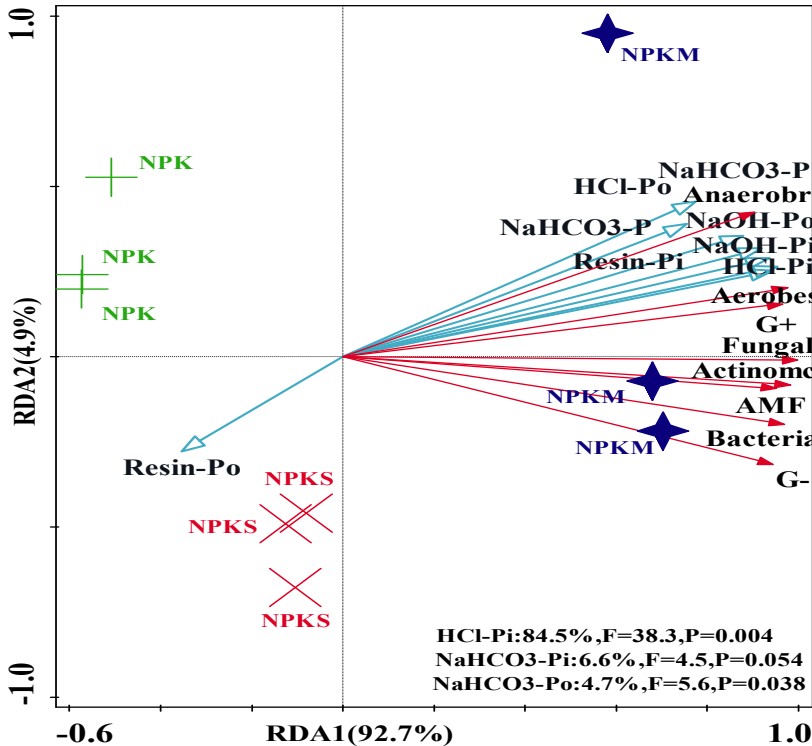

**Figure 5.** Redundancy analysis was performed on soil microbial community structure marked by PLFA profiles and P fractionations by chemical sequential fractionation under different long−term fertilization.

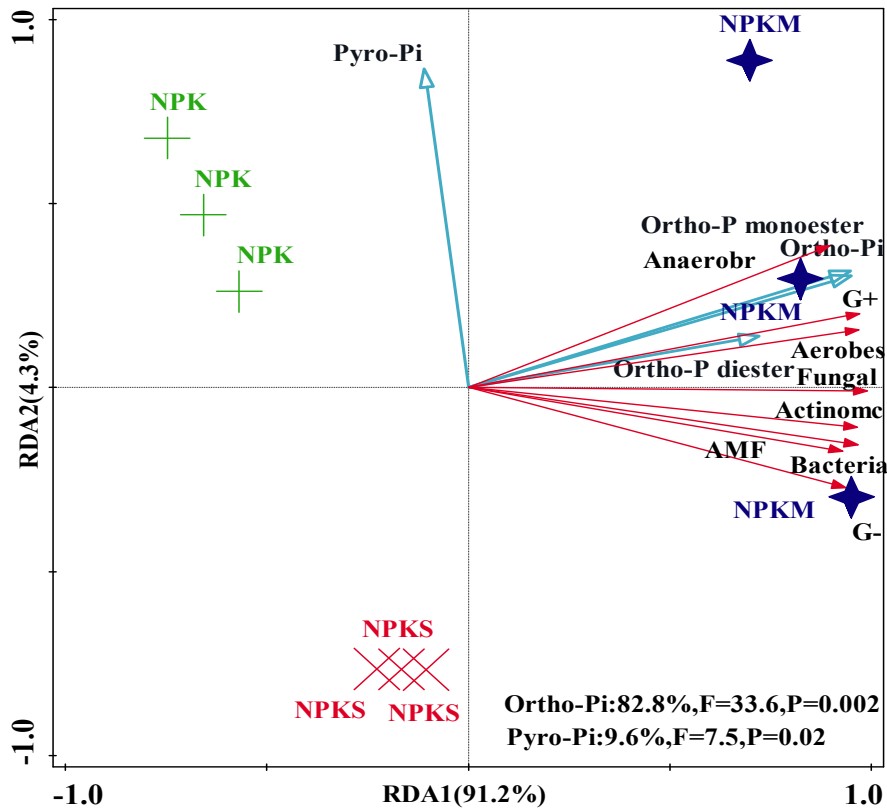

**Figure 6.** Redundancy analysis was performed on soil microbial community structure marked by PLFA profiles and P species by $^{31}$P-NMR spectroscopy under different long−term fertilization.

## 4. Discussion

Long-term application of manure and straw (NPKS and NPKM) greatly affected pH, TC, TN, and P content including TP, Olsen-P, and Po. NPKM and NPKS treatments resulted in an increase of soil pH compared with NPK. It is obvious that soil pH became lower with NPK application. Long-term application of chemical fertilizer with more ammonium would result in an increase in soil acidity [37]. Some studies indicated that application of manure and straw could alleviate soil acidification [38,39]. Soil organic matter can complex acid ions or release exchangeable cations [38]. NPKM-treated soil showed the highest values of total C and N contents compared with NPK and NPKS, which possibly accounted for the fact that manure contained higher C and N contents.

The present study showed that long-term application of NPKM enhanced the concentrations of soil total P, Po, and Olsen-P compared with NPK and NPKS. Through the $^{31}$P-NMR experiment, we detected orthophosphate, orthophosphate monoesters, and pyrophosphate in the three treated soils. Orthophosphate and phosphate monoesters accounted for most of the total extracted P amounts. This consistent result was reported by Xin et al. (2019) [29] and Wei et al. (2017) [40]. Compared with NPK and NPKS, long-time NPKM application notably increased the contents of orthophosphate and phosphate monoesters. However, the proportions of phosphate monoesters in total extracted P amounts declined as the contents of orthophosphate increased under NPKM treatment. Annaheim et al. (2015) [27] also found that orthophosphate was the major form of total extractable P, accounting for 68–91%, in the topsoils with 62 years of application of dairy manure, compost, and dried sewage sludge. Koopmans et al. (2007) [24] indicated orthophosphate and orthophosphate monoesters accumulated in heavily manured topsoils in the Netherlands. Orthophosphate analyzed using $^{31}$P-NMR spectroscopy was found to increase with increasing additions of mineral P fertilizer alone or together with manure in long-term field experiments in Sweden [41]. IP is a phosphorous compound in an organic binding state existing in plant seeds and germ. It is generated by replacing six hydroxyl

groups with phosphate groups on inositol. IP contains nine isomers, including seven non-optically active forms and two optically active forms [42]. Compared with the NPK treatment, NPKM application significantly increased P species. Shown in Figure 2(c1–c4) were the characteristics peaks of *myo*-IP, which cannot be clearly identified in the $^{31}$P-NMR spectrum as signals overlapped with other Po compounds in the soil extract with NaOH-EDTA solution. Phosphorylethanolamine was a precursor of phosphorylcholine, which was an important product during plant growth and a precursor of synthesis phosphatidyl-choline in the cytomembrane [43]. A large proportion of Pi and *myo*-IP presented in manure, and this indicated that manure application tended to increase Pi and *myo*-IP concentrations in soil [29]. The majority of P in poultry litter extracted using NaOH and HCl solution was in Po forms, which was easily subjected to enzymatic hydrolysis in soil [44]. Po mineraliza-tion is affected by soil C/P ratio. Under C/P less than 200, Po would be easily mineralized. The value of C/P in manure was generally lower than 50, and most Po in manure applied to soil would be readily transformed to Pi [45].

Sequential P fractionation method is commonly used to assess the availabilities of various P forms and to detect the P transformation in soils based on evaluating how tightly the different P fractionations are bound to the soil matrix [2,7,8]. The present study used a modified version of the Hedley fractionation scheme developed by Guppy et al. (2000) [10] to determine P fractions in soil as influenced by long-term application of NPK, NPKS, and NPKM. Our result showed that resin-P and NaHCO$_3$-P accounted for 19~32% of the total P, and HCl-P was 36~39%, followed by NaOH-P, 22~30%. The P fraction in the NaOH extract was found to be associated with iron (Fe) and aluminum (Al) oxides in soils, and the P form in the HCl extract was considered to bound to calcium (Ca) [11]. Resin P and NaHCO$_3$-P are regarded as labile P, NaOH-P is moderately labile P, and HCl-P and residual P as non-labile or stable P pools based on their availability in soils for plant utilization [7,8]. HCl-Pi could be partially utilized by plant roots with a strong ability to secrete H$^+$ ion and organic acids [46,47]. Residual P is difficult to decompose or dissolute and consists mainly of passive phosphates in the soils [12]. NPKM application notably increased Resin-Pi, NaHCO$_3$-Pi and Po, NaOH-Pi and Po, HCl-Pi and Po. There was no significant difference in concentrations of soil various P forms between NPK and NPKS treatments. It is concluded that long-term application of NPKM obviously enhanced the concentrations of labile P, and moderately labile P in mollisol. This conclusion was consistent with the result of Saleque et al. (2004) [48]. They also found that the addition of cow manure markedly enhanced soil labile, NaOH-Pi, and HCl-P pools. Excessive P application in pomelo orchard soil showed a remarkable influence on the soil P fractions. The application at a rate of 905.4 kg P$_2$O$_5$ ha$^{-1}$ each year resulted in high P surplus in soil and increases of the contents of Fe- and Al-bound P [49].

Microbial growth and activity in soils were promoted by manure and chemical fertil-izer application for 35 years [40]. Microorganisms decompose organic matter and release more P to soil, especially under the manure application. RNA, DNA, and PLFAs are P-containing compounds in the cells of soil microorganisms [50,51]. RNA content varied with bacteria growth rate [52]. Under alkaline conditions, the nucleotide monomer generated from RNA hydrolysis easily occurred [53,54]. The fatty acid and polar head groups of phospholipids might convert to α- and β-glycerophosphate [55,56]. α-glycerophosphate was positively correlated with soil microbial biomass P [34,57]. Our study found that NPKM application significantly increased the species of orthophosphate esters in soil, such as β-glycerophosphate and sugar phosphates or other unknown monoesters and other mononucleotides from degradation products of RNA (Figure 2).

PLFAs are specific components of cell membranes in living microorganisms [21]. Vari-ous microorganisms form different kinds of PLFAs through different synthetic pathways and biochemical processes [58]. We found NPKM application increased the amount of soil total PLFAs and the PLFAs contents in bacteria, G$^+$, G$^-$, actinomycetes, and fungi, sug-gesting that NPKM increased soil microbial functional diversity. Qaswar et al. (2021) [59] also indicated long-term fertilization exerted a notable influence on total PLFA contents,

suggesting soil microbial community structure was changed. Compared with the NPK and no fertilizer treatments, combined application NPK with manure lead to significant increases of soil total C, N, P, and PLFA concentrations in the layer deep 20 cm of the paddy soil. Our PCA results also showed that soil microbial community composition was separated via PCA mainly on PC1 (92.7%), and NPKM treatment was at the positive end of PC1 (Figure 3), suggesting significant differences of soil microbial community structure occurred among the treatments of NPK, NPKS, and NPKM. The loading values of PCA indicated that PLFA biomarkers were most important for the NPKM treatment, suggesting the abundance of soil microbial community composition was the highest in this treatment (Figure 3). Zhang et al. (2015) [60] and Tian et al. (2017) [61] indicated that manure addition can provide stable and labile substrate for supporting microbial rapid growth, which might result in an increase of total PLFAs in soils. The proportions (%) of $G^-$ bacteria in total PLFAs increased under straw application, while the proportions (%) of $G^+$ bacteria significantly increased under manure treatment. $G^-$ bacteria tend to use organic materials with a low degree of humification, while $G^+$ bacteria tend to use organic materials with a high humification degree [62].

Long-term fertilization of manure directly input nutrient to soils and change of soil C/N ratio, which might have impact on the soil microbial community [63,64]. Some research showed that fertilization regimes exerted the greatest effects on bacterial and fungal community structures, and NP fertilizers tended to inhibit arbuscular mycorrhizae [65]. Pan et al. (2020) [66] found soil bacterial communities were higher under long-term application of manure than inorganic fertilizers, and fungal communities showed the opposite trend. It has been estimated that 1–50% of bacteria and 0.1–0.5% of fungi in soil can carry out P solubilization. *Bacillus* is the most predominant genus among P-solubilizing bacteria [67]. Bi et al. (2020) [68] suggested that soil microbes could facilitate P cycle by enhancing the hydrolysis of soil Po compounds and promote the turnover of labile or relative available P fractions such as $H_2O$-Pi, $NaHCO_3$-Pi, and NaOH-Pi in soil. Wang et al. (2020) [69] found that soil pH and organic C had significant impact on the soil microbial community, which explained 51% and 20% of the variation in the non-rhizosphere of in Chinese Hapludults soil, respectively. However, only soil pH remarkably affected microbial community structure in the rhizosphere, which explained 31% of the variation. In our study, the RDA results displayed that microbial community composition was greatly influenced by soil pH, TC (Figure 4), HCl-Pi, and $NaHCO_3$-Po (Figure 5). We also found that the content of Pi including orthophosphate and pyrophosphate identified by the $^{31}P$-NMR experiment were related to the PLFAs of anaerobe, aerobes, and $G^+$. There was a close correlation between HCl-Pi, $NaHCO_3$-Po, orthophosphate, and pyrophosphate with anaerobe, aerobes, and $G^+$. The relationship between the transformation of Po to Pi with soil microbial community such as anaerobe, aerobes, and $G^+$ in soils with long-term fertilization is an interesting aspect that deserves further investigation. In summary, long-term application of NPKM obviously promoted the diversity of the soil microbial community in mollisol, which might be associated with changes of soil pH, total carbon, and labile P forms under manure application.

## 5. Conclusions

The phosphorus in soil was mainly HCl-P, followed by NaOH-P. NPKS and NPKM increased soil pH, TC, and TN contents. NPKS application did not change soil phosphorus form, but NPKM application increased Resin P, $NaHCO_3$-P, NaOH-P, and HCl-P. Compared with NPK, both NPKS and NPKM changed soil microbial community composition. The number of $G^+$, $G^-$, fungi, and actinomycetes in NPKM increased significantly. The changes of soil TC, pH, HCl-Pi, and $NaHCO_3$-Po were the main contributing factors to the changes of the microbial community structure. Phosphodiester conversed to monophosphate and orthophosphate under the action of microorganisms. Therefore, long-term manure fertilization with chemical fertilizer application is a good measure to improve soil phosphorus pool

in mollisol and microbial community diversity and also has an obvious effect on increasing yield of maize.

**Author Contributions:** The manuscript was completed through the contributions of all authors. H.W. organized and designed the experiments. S.S., J.Z. and Y.L. conducted the experiments. S.S. and H.W. wrote the manuscript and participated in the discussion and revision of the manuscript. All authors have read and agreed to the published version of the manuscript.

**Funding:** This work was supported by the National Key Research and Development Program of China (2016YFD0200108), National Basic Research Program of China (973 Program) (2013CB127402) and Hainan Provincial Natural Science Foundation of China (320QN324, 322MS119).

**Data Availability Statement:** Not applicable.

**Acknowledgments:** The authors are grateful to Bruker BioSpin offices in China and Institute of Process Engineering, Chinese Academy of Sciences for providing sample determination of NMR experiment. This work was supported by the National Key Research and Development Program of China (2016YFD0200108), National Basic Research Program of China (973 Program) (2013CB127402) and Hainan Provincial Natural Science Foundation of China (320QN324, 322MS119).

**Conflicts of Interest:** The authors declare that they have no conflict of interest.

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
