# Peer review of "Accumulation of Labile P Forms and Promotion of Microbial Community Diversity in Mollisol with Long-Term Manure Fertilization"

_agronomy, doi:10.3390/agronomy13030884_

Round 1

Reviewer 1 Report

The authors of the peer-reviewed manuscript undertook a comprehensive study of soil properties (chemical and microbiological) after 26 years of continuous maize monoculture in Northeast China. The soil was subjected to 3 types of fertilization: chemical fertilization (NPK), NPK with cereal straw (NPKS) and NPK with manure (NPKM). I believe that the doses of individual fertilizers were set correctly, and the entire agrotechnics related to the methods of fertilization does not raise any objections. The most advantageous fertilization variant turned out to be the long-term use of mineral fertilization with manure (NPKM). This variant promoted the diversity of soil microbial communities and led to the accumulation of labile and intermediate labile P in mollisol. Compared to NPK, both NPKS and NPKM changed the composition of the soil microbial community. The number of G+ , G- , fungi and actinomycetes in NPKM increased significantly. This, in turn, influenced the effective conversion of phosphodiesters into monophosphates and orthophosphates available for maize plants.

I find the conducted research interesting and necessary. It is a pity that such a comprehensive soil study was not carried out in other time periods than the years 1990 - 2016, e.g. in 2000, 2010?. Corn is a monoculture-resistant crop, but monoculture of this crop lasting 26 years is not that common. It would be interesting to see whether, for example, after 10 or 15 years of monoculture, the impact of individual fertilization variants would be similar to that after 26 years, or different (better/worse). Moreover, in the initial year of cultivation, only the basic parameters of soil chemical properties were examined. The other comprehensive chemical and microbiological soil properties studied in 2016 cannot be compared over time. I suggest that the authors refer to this issue in "Discussions" or "Conclusions". The above remark is only my suggestion and does not diminish the high substantive and practical value of the manuscript. Therefore, you may consider accepting the article for publication in its present form.

06.01.2023.

Reviewer 2 Report

The changes of soil P forms and associated microbial community composition in mollisol with long-term fertilization was studied. The manuscript is written well and the research method is reliable. However, the manuscript is lack of innovation and illustrate the significance and importance of this study is suggested, especially in discussion section, which need a thorough discussion about the link between P and microbes. After the following comments address, the manuscript could be further considered.  

General comments:

1.      In the abstract, the author only describes the results about P forms and microbial communities under different treatment. However, is there any linkage between P behavior and microbes? The two mainly results should be linked rather than parallel. Moreover, the significance and importance of this study also need to be clarified.

2.      There have been many studies on the effects of organic fertilizer application on soil phosphorus forms and microbial community changes. Compared with other relevant studies, what is the innovation in this paper. Please indicate in the manuscript.

3.      In this study, Soil P forms were analyzed by a chemical sequential fractionation method and solution 31P nuclear magnetic resonance technique, but only the labile P forms are presented in the title. Moreover, no related information in the manuscript that which forms are included in labile and stable P forms. Please give an explanation.

4.      Although the author briefly introduced the current research progress in the research background, too much information about P forms methods, please make this section concisely. Moreover, why you focus on soil P forms and microbes in response to long term straw and manure incorporation? The introduction should be reorganized.  

Specific comments:

5.      Line 241, please delete “Markley”.

6.      It is suggested to reorganize the format of Table 3, which is too messy.

7.      Figure 4 - Figure 6, it is hard to distinguish that which arrow corresponding to the letters.

8.      There are many factors that could change the microbial community in the soil environment. For example, straw incorporation may affect soil P by changing soil pH, or through the adsorption and desorption processes. How can you effectively explain that the change of microbial community caused by soil phosphorus pool?

9.      Is there any basis for the amount of manure and straw application in this study? The addition amount of manure or straw is not equal in terms of the amount of carbon. The amount of manure is higher than straw, so is it reasonable to analyze the significance of adding manure and straw?

Reviewer 3 Report

Dear Authors,

I revised the manuscript  Accumulation of labile P forms and promotion of microbial community diversity in mollisol with long-term manure fertilization   submitted to the Agronomy Journal.

 The subject of the experiments has been carried out for many years and in many countries.  The publication documents and describes well the changes in soil phosphorus content and microbial changes under the influence of NPK, NPKS and NPKM fertilisation.

The paper is very interesting. However, I have some concerns, which need to be addressed before considering for final publication.

 1. INTRODUCTION

The authors could have paid more attention in the literature review to the effect of pH (too acidic or alkaline) on the availability of phosphorus in the soil for plants. This is still a rather serious problem under European conditions.

 2.1. Filed experiment

Question:

Line 145.  In which phase of maize growth second third of N was used?

Request to complete in the text what was the annual mean precipitation mm and mean temperature oC of the multi-year period from 1990 to 2016.

During the conduct of this experiment, grain maize varieties with what FAO were used ?

2.2. Soil samples and chemical analysis

Research methods used correct, I have no objections.

2.3. NMR experiments

2.4. Sequential fractionation

2.5. Soil PLFA analysis

The research methods used and the choice of measuring equipment correct, I have no objections.

 Line 231  - Table 1.

Can the authors complete the resultant commentary as to what may have influenced the lower maize grain yield in the NPKS variant (11464.3 kg-ha-1) compared to NPK application alone (11789.1 kg-ha-1) ? Many of the determined result parameters were higher with NPKS compared to NPK application alone.

What then influenced the lower yields of the NPKS variant? In this variant, additional organic matter in the form of straw was introduced.

 Results presented in tables and figures legible and clear.

Tables 1 and Figures 1,3,4,5,6

Please do not duplicate the entire records of fertilisation options under Tables 1 and Figures 1,3,4,5,6 (NPK: mineral fertilizers with nitrogen, phosphorus, and potassium; NPKS: straw at 7.5 t ·ha-1+NPK; NPKM: manure at 30 t ·ha-1+NPK.). NPK, NPKS and NPKM alone are readable to the viewer. A detailed description of fertilisation is already in the methodology in lines 135-140.

 Observation and analysis of results correct.

Conclusions of the analysis correct. I have no major objections to the analyses carried out and the results obtained. They also confirm the results obtained by other authors of similar experiments.

 The subject of the influence of NPK fertilisation, the use of straw for fertilisation and, in particular, the use of straw, is not new, but the publication has been extensively analysed and a great deal of work has been done. The results obtained are interesting to use and cite in other works of this type as confirmation of the usefulness of fertilisation with manure of agricultural crops, in particular maize monoculture or other cereals.

 Once the Authors have made additions and the comments of the other reviewers have been taken into account, the publication will constitute interesting scientific material.

Editorial corrections concerning the list of literature used are left to the Editor.

Congratulations to the entire team of Authors of the paper: Shuhui Song, Jinyao Zhang, Yunxia Liu and Hong Wang for their perseverance, consistency in analysis and diligence throughout the research.

Round 2

Reviewer 2 Report

Authors have made the corrections.